# A Novel Universal Primer Multiplex Real-Time PCR (UP-M-rtPCR) Approach for Specific Identification and Quantitation of Cat, Dog, Fox, and Mink Fractions Using Nuclear DNA Sequences

**DOI:** 10.3390/foods12030594

**Published:** 2023-01-31

**Authors:** Wenjun Wang, Tiean Wei, Manna Shi, Yu Han, Yang Shen, Xiang Zhou, Bang Liu

**Affiliations:** 1Key Laboratory of Agricultural Animal Genetics, Breeding, and Reproduction of Ministry of Education, Hongshan Laboratory, Huazhong Agricultural University, Wuhan 430070, China; 2College of Animal Science and Technology, Hebei Agricultural University, Baoding 071000, China; 3Hubei Hongshan Laboratory, Wuhan 430070, China

**Keywords:** carnivorous animals, nuclear DNA target sequences, universal primers multiplex real-time PCR (UP-M-rtPCR), quantitative detection, meat adulteration

## Abstract

Adulteration of meat with carnivorous animals (such as cats, dogs, foxes, and minks) can cause ethical problems and lead to disease transmission; however, DNA quantitative methods for four carnivorous species in one tube reaction are still rare. In this study, a carnivore-specific nuclear DNA sequence that is conserved in carnivorous animals but has base differences within the sequence was used to design universal primers for its conserved region and corresponding species-specific probes for the hypervariable region. A novel universal primer multiplex real-time PCR (UP-M-rtPCR) approach was developed for the specific identification and quantitation of cat, dog, fox, and mink fractions in a single reaction, with a 0.05 ng absolute limit of detection (LOD) and 0.05% relative LOD. This approach simplifies the PCR system and improves the efficiency of simultaneous identification of multiple animal-derived ingredients in meat. UP-M-rtPCR showed good accuracy (0.48–7.04% relative deviation) and precision (1.42–13.78% relative standard deviation) for quantitative analysis of cat, dog, fox, and mink DNA as well as excellent applicability for the evaluation of meat samples.

## 1. Introduction

Some carnivorous animals, such as cats, dogs, foxes, and minks, are potential carriers of zoonotic diseases (e.g., trichinellosis, leishmaniasis, and rabies) [1,2] and may even carry and spread SARS-CoV-2 [3,4]. Exposure to, or consumption of, products from these animals can cause serious health problems [5,6]. Additionally, their meat is prohibited by Muslim dietary laws and is usually defined as inedible in most countries. Cats and dogs are the most common and popular companion pets [7], and consuming their meat is unbearable and unethical for most people. As fur animals, foxes and minks contain large amounts of heavy metal residues in their meat and are not recommended for consumption [8,9]. However, some unscrupulous merchants use the carcasses of foxes and minks, as well as hunted stray cats and dogs, to substitute beef or mutton for sale [10,11]. Adulteration with cat, dog, fox, or mink meat may not only cause religious and economic issues but also raises serious health problems. To prevent carnivorous meat from being illegally mixed with other meats for sale, an accurate and efficient method is urgently needed to identify these animal-derived ingredients.

Numerous methods have been developed for detecting meat adulteration based on protein, RNA, and DNA analyses [12,13]. Among them, DNA is considered the most suitable biomarker for species identification because of its high stability during thermal processing and sufficient variation among different species [14]. A variety of DNA-based approaches for meat species authentication are widely recognised, including species-specific PCR [15], PCR-RFLP [16], real-time PCR [17], and droplet digital PCR [18], which play vital roles in preventing meat adulteration. Since multiplex PCR methods are more time-saving, efficient, and economical than simplex PCR methods, more researchers will prefer to use multiplex PCR methods to authenticate multiple target species in the future [19]. However, conventional multiplex PCR methods based on amplicon length [20], fluorescent signal [21], or melting curve analysis [22] require the design of specific primers to detect different meat species simultaneously. An increase in the number of primers increases the complexity of the reaction system. Primer interactions and different amplification efficiencies may have a significant impact on the experimental results and are the major limiting factors for the wide application of multiplex PCR methods [23]. While ensuring the specificity of each primer pair, their melting temperature should also be considered, which is a significant challenge for non-experienced researchers.

Advances in DNA technology have led to rapid development of alternative methods for species identification. Some detection methods using universal primers combined with electrophoresis or sequencing have been established in recent years to address these problems [24,25,26]. Unfortunately, these methods are often time-consuming, complicated, expensive, and difficult to distinguish between closely related species, which severely limits their widespread application. Additionally, the majority of these methods focus on economically valuable edible meats, such as cattle, sheep, and pigs [27,28] but rarely on cats, dogs, foxes, and minks. The lack of available target sequences to simultaneously identify cats, dogs, foxes, and minks is another factor that restricts the development of analytical methods. Mitochondrial DNA (mtDNA) is often the preferred target for species identification due to the good sensitivity resulting from high copy number [29]. However, mtDNA has higher variation and copy number differences across breeds, individuals, and tissues than nuclear DNA has [30,31]. This may result in false-negative results, making it unsuitable for accurate quantification [32]. In contrast, nuclear DNA is relatively conserved and stable in different species and tissues and is better suited for quantitative analysis of meat adulteration. Generally, an appropriate nuclear DNA target sequence for designing universal primers should meet the following requirements: the ends of the sequence should be highly conserved, with a few base differences in the sequence and preferably no homologous sequence with the genomes of other non-target species [33].

The aim of this study was to screen a nuclear DNA sequence for the specific detection of cat, dog, fox, and mink DNA. By designing universal primers in conserved regions and species-specific probes in the hypervariable regions of the nuclear DNA sequence, a simple, accurate, and efficient universal primer multiplex real-time PCR (UP-M-rtPCR) approach was developed for the simultaneous identification and quantification of cat, dog, fox, and mink fractions in meat products. The UP-M-rtPCR method will be conducive in simplifying the reaction system, improving detection efficiency, and providing technical support to ensure the safety of meat products.

## 2. Materials and Methods

### 2.1. Sample Collection

Raw meat samples from 21 species were used as experimental material. Cat and dog samples were provided by the Veterinary Hospital of Huazhong Agricultural University (China, animal experiment approval Nos. HZAUCA-2020-001, and HZAUDO-2020-002, 19 August 2020, Ethical Committee of Huazhong Agricultural University). Fox and mink samples were collected from a fur-bearing animal-breeding farm in Hebei Province, China. Mouse, rat, hamster, and guinea pig samples were obtained from the Laboratory Animal Center of Huazhong Agricultural University (animal experiment approval Nos. HZAUMO-2020-0089, HZAURA-2020-0007, HZAUGP-2020-001, and HZAUGP-2020-002, 17 August 2020, Ethical Committee of Huazhong Agricultural University). Meat samples of other species including chicken, duck, goose, domestic horse, donkey, yak, cattle, buffalo, goat, sheep, pig, deer, and rabbit were purchased from local markets. All samples were stored at −80 ℃ until use.

### 2.2. Preparation of Meat Samples

To determine the relative limit of detection (LOD), four types of binary meat mixtures (cat/cattle, dog/cattle, fox/cattle, and mink/cattle) with a total weight of 100 g were prepared by mixing 5%, 0.5%, 0.05%, and 0.005% meat of each target species (cat, dog, fox, and mink) with beef. To evaluate the applicability and accuracy of our method, six artificial meat samples for mimicking adulteration were prepared by mixing the meat of multiple target species with beef or mutton. The meat mixtures were minced and homogenised using a blender (Sunbeam Oster, Poca Raton, FL, USA).

### 2.3. DNA Extraction

Genomic DNA was extracted from 50 mg of pure meat or a meat mixture using standard SDS/proteinase K and phenol/chloroform treatments, including SDS/proteinase K digestion, phenol:chloroform extraction, and ethanol precipitation steps [34]. The extract was further purified by MicorElute DNA Clean-Up Kit (Omega Bio-tek, Doraville, GA, USA). The purity and concentration of the extracted DNA were determined using a Nanodrop 2000 spectrophotometer (Thermo Fisher Scientific, Wilmington, DE, USA). All DNA samples were diluted to 50 ng/µL for PCR.

### 2.4. Screening Nuclear DNA Target Sequences of Cat, Dog, Fox, and Mink

The nuclear DNA sequences of dogs were aligned with the genome sequences of pigs, cattle, sheep, horses, rabbits, mice, chickens, and ducks downloaded from GenBank using the local BLAST+ tool (NCBI BLAST+ 2.12.0+). The sequences with the alignment parameters of ‘Query-length’ >200 bp, ‘Identity’ <80%, and ‘E-value’ <10^−5^ were selected and aligned with the genome sequences of cat, fox, and mink, and the sequences with ‘Query-length’ >200 bp and ‘Identity’ >80% were identified as the candidate target sequences. Subsequently, these sequences were aligned with the reference genome sequences of 40 other species, including representative species of four kingdoms, 10 classes, 19 orders, 28 families, and 37 genera (ostrich, goose, duck, turkey, chicken, human, hedgehog, elephant, pig, deer, horse, donkey, camel, sheep, goat, zebu, buffalo, cattle, yak, pika, rabbit, squirrel, rat, mouse, guinea pig, beaver, hamster, jerboa, fruit fly, salmon, zebrafish, frog, anole, wheat, rice, maize, soybean, *Salmonella*, *Escherichia coli*, and yeast), using online nucleotide BLAST (https://blast.ncbi.nlm.nih.gov/Blast.cgi. accessed on 10 September 2020). Homologous sequences were retrieved and aligned using ClustalW multiple sequence alignment (https://www.genome.jp/tools-bin/clustalw. accessed on 15 September 2020) to determine the conserved and hypervariable regions. Sequences with relatively low homology with non-target species, high conservation at the two ends, and sufficient base differences in the middle of the target sequence were selected as nuclear DNA target sequences.

### 2.5. Design of Universal Primers and Species-Specific Probes

Universal primer pairs were designed based on the rules of universal primer designing for the relatively conserved region of the nuclear DNA target sequence using Primer-BLAST (https://www.ncbi.nlm.nih.gov/tools/primer-blast/, accessed on 9 October 2020). The corresponding species-specific probes were designed for the hypervariable region for the differentiation of cats, dogs, foxes, and minks. Detailed information on the universal primer pairs and species-specific probes is presented in Table 1.

### 2.6. Conventional PCR

The conventional PCR assay was performed using a Bio-Rad MyCycler^TM^ Thermal cycler with a 20 µL total reaction volume composed of 50 ng DNA template, 200 nM of each primer, 0.5 U TaKaRa Taq polymerase, 1.6 µL of dNTP mixture, 1× PCR buffer (Mg^2+^ Plus), and sterile water. The cycling parameters were initial denaturation at 95 °C for 3 min, followed by 35 cycles of denaturation at 95 °C for 30 s, annealing at 58 °C for 30 s, elongation at 72 °C for 20 s, and a final elongation at 72 °C for 2 min. The PCR products were visualised on a 2% agarose gel stained with GelRed (Biotium, Fremont, CA, USA) in 1 × Tris-acetate-EDTA buffer and analysed using a GelDoc 1000 gel Documentation System (Bio-Rad Laboratories, Inc., Hercules, CA, USA).

### 2.7. Real-Time PCR

Simplex real-time PCR was performed using a total reaction volume of 20 µL containing 50 ng DNA template, 1 × Premix Ex Taq™ (Probe qPCR) (Takara Bio, Beijing, China), 200 nM of each primer, 300 nM probe, and sterile water. Multiplex real-time PCR was performed using a total reaction volume of 20 μL containing 50 ng mixed DNA of cat, dog, fox, and mink; 1× Premix Ex Taq™ (Probe qPCR) (Takara Bio); four probes (300 nM Cat-P, 150 nM Dog-P, 400 nM Fox-P, and 100 nM Mink-P); and sterile water. Both simplex and multiplex real-time PCR were performed using a CFX96™ Touch Real-Time System (Bio-Rad Laboratories, Inc.) using the optimal cycling conditions: 95 °C for 3 min, followed by 40 cycles at 95 °C for 10 s and 59 °C for 30 s. Each sample was analysed three times in parallel. Non-template controls were included in each experiment to monitor contamination.

### 2.8. Construction of Standard Curves for Cats, Dogs, Foxes, and Minks

Three types of standard curves for cats, dogs, foxes, and minks were constructed. First, the genomic DNA of the cat, dog, fox, and mink was serially diluted with double-distilled water to 5, 0.5, 0.05, and 0.005 ng/μL as templates for UP-M-rtPCR. The Cq values were plotted against the logarithms of the DNA concentrations to create standard curves for each target species. Second, another type of standard curve for cats, dogs, foxes, and minks was constructed by plotting different weight proportions (5%, 0.5%, 0.05%, and 0.005% *w*/*w*) of each target species in binary meat mixtures vs. Cq values. The Cq values were plotted against the logarithms of the DNA concentrations to create standard curves for each target species. Finally, 50 ng/µL genomic DNA from cats, dogs, foxes, and minks was serially diluted with cattle DNA to different proportions (1%, 5%, 10%, 50%, and 100%) as the template for UP-M-rtPCR. Four standard curves were generated and used for quantitative detection of the DNA proportions of cats, dogs, foxes, and minks. The formulae of the standard curves and their R^2^ values were calculated.

### 2.9. Quantitative Analysis of DNA Mixtures

To verify the accuracy and precision of the UP-M-rtPCR method, three DNA mixtures with varying proportions were used to quantitatively analyse the cat, dog, fox, and mink fractions. Combined with the standard curve formulae of the four target animals, the percentage of cat, dog, fox, and mink DNA was calculated by analysing the real-time PCR data. The formula for the standard curve is *y* = *a* x + *b*, and the process for calculating the DNA proportion is as follows:$$C=10^{\frac{y-b}{a}}\times 100\%$$
where C represents the percentage of cat, dog, fox, or mink DNA; *y* is the Ct value obtained from the UP-M-rtPCR of a particular sample; and *a* and *b* are the slope and intercept of the corresponding standard curve formula, respectively.

## 3. Results and Discussion

### 3.1. Identification of Nuclear DNA Target Sequence, Universal Primers, and Species-Specific Probes for Cat, Dog, Fox, and Mink DNA

Appropriate biomarkers are required for the identification of various animal-derived ingredients in meat products. In this study, a DNA sequence from the dog nuclear genome region (NC_051821.1) was screened as the target sequence for specific identification and differentiation of cats, dogs, foxes, and minks. The results of the nuclear DNA target sequences aligned with those of 40 other non-target species, as well as the designed regions of primers and probes, are shown in Figure 1. Homologous regions of the target sequences were retrieved for only a few non-target species genomes, implying that this sequence is highly specific in carnivorous animals. The sequence homology of cat, dog, fox, and mink was significantly higher than that of other non-target species, especially in the primer-binding regions. This enabled simultaneous and specific detection of the four species using only one pair of primers. Generally, a key mismatch in the primer-binding region may result in reduced amplification efficiency (AE) and even false-negative results [35]. The potentially unequal amplification of multiple targets in the system is a major disadvantage of conventional multiplex PCR methods [36]. Our study designed a pair of universal primers for the relatively conserved sequence region of the four target species that did not bind to the sequences of non-target species. The single nucleotide polymorphisms in the universal primers were replaced by degenerate bases so that different targets had identical AE and no one target was amplified in preference to another. Additionally, different fluorescently labelled probes were designed in the hypervariable regions of cat, dog, fox, and mink target sequences to quickly distinguish them and to greatly improve detection accuracy (Table 1). At least four base differences in the probe-binding region of different target species were found, and the sequence difference of non-target species in this region was more obvious than that of each probe. Therefore, nuclear DNA target sequences, universal primers, and species-specific probes are well-suited for identifying and distinguishing cats, dogs, foxes, and minks.

### 3.2. Evaluation of Specificity and Conservation of the Nuclear DNA Target Sequences, Primers, and Probes

The specificity of target sequences, primers, and probes is a crucial prerequisite for species identification [37,38]. In preliminary experiments, conventional PCR was performed to evaluate the specificity of universal primers (Figure 2A). The universal primers amplified the expected 339, 335, 335, and 332 bp fragments from cat, dog, fox, and mink DNA, respectively. No specific amplified products were observed for other non-target species. These results showed that the universal primers had high specificity to the nuclear DNA target sequences and could be used in subsequent species identification experiments. Furthermore, PCR results of five individual samples from each species showed that the target sequence and primers exhibited low heterogeneity, without major sequence variation in the amplified regions (Figure 2B). Next, simplex real-time PCR was performed to verify the specificity of each probe for cats, dogs, foxes, and minks (Figure 2C). A significant fluorescent signal was only detected from the corresponding target species DNA, indicating that the probes had sufficient ability to distinguish cats, dogs, foxes, and minks. Moreover, the sequencing results of PCR products from different individuals also confirmed that the nuclear DNA target sequences of each target species were consistent with the expected sequence shown in Figure 1, especially in the binding region of primers and probes, which can effectively prevent the occurrence of false negatives.

### 3.3. Establishment and Optimisation of UP-M-rtPCR for the Detection of Cat, Dog, Fox, and Mink Components

In a multiplex real-time PCR system, the ratio of the probes affects the specificity and sensitivity of the reaction [39]. The proportion of each probe was optimised to ensure significant fluorescence signals and the lowest Cq values for the simultaneous detection of cat, dog, fox, and mink components by real-time PCR. Optimal results were obtained when the probe ratios of cats, dogs, foxes, and minks were 6:3:8:2. Considering that the interaction between universal primers and probes can decrease detection accuracy, it is necessary to test the specificity of the UP-M-rtPCR system. Cross-specificity was tested by UP-M-rtPCR in 21 species, including four target species (cat, dog, fox, and mink) and 17 non-target species (domestic horse, donkey, yak, cattle, buffalo, goat, sheep, pig, deer, rabbit, mouse, rat, hamster, guinea pig, chicken, duck, and goose). As shown in Figure 3A, a significant fluorescence signal was generated only when the target DNA was detected. This clearly demonstrates that no cross-reaction with any non-target animal-derived ingredients was detected until the PCR cycles exceeded 36, and a Cq value of 36 was used as the cut-off point. Furthermore, the four independent fluorescence amplification curves in the UP-M-rtPCR assay showed that it could be used to detect cats, dogs, foxes, and minks simultaneously in one reaction (Figure 3B). In conclusion, the UP-M-rtPCR system showed good specificity for 21 species. The four target species were identified and differentiated in a single reaction. Since many meat products may be unintentionally or incidentally mixed with several target species, multiplex PCR for simultaneous detection can save considerable time and labour costs [40]. However, the sensitivity and accuracy of conventional multiplex PCR methods are inevitably reduced owing to the potential cross-reactivity between multiple primer pairs [41]. In this study, a multiplex real-time PCR approach using only one pair of universal primers, named UP-M-rtPCR, was developed, which can simplify the reaction system and fundamentally eliminate or reduce the probability of potential primer–primer/primer–probe dimers. The UP-M-rtPCR approach can specifically identify and distinguish cats, dogs, foxes, and minks in a reaction in approximately one hour, thereby improving the detection efficiency and accuracy along with saving time and reducing the cost of the reaction.

### 3.4. Sensitivity of UP-M-rtPCR

Sensitivity is an important indicator for evaluating the performance of a new method. The sensitivity of UP-M-rtPCR was evaluated using two indices: absolute and relative LOD. First, the genomic DNA of cat, dog, fox, and mink was serially diluted 10-fold from 5 to 0.005 ng and used as a template to determine the absolute LOD. An obvious fluorescence signal was stably detected from as low as 0.05 ng of the target species DNA (Cq value < 36) (Figure 4A), which showed that our method provided high sensitivity while simplifying the reaction system. The Cq values decreased linearly with the logarithm of the DNA concentration for each target species (Figure 4B). The correlation coefficients (R^2^) for cat, dog, fox, and mink DNA were 0.9914, 0.9992, 0.9880, and 0.9887, respectively, which met the acceptance criteria of ENGL [42] (R^2^ ≥ 0.98). As shown in Appendix A, the standard deviation (SD) of the measured Cq values in the three replicates was 0.03–0.32, and the coefficient of variation (CV) was 0.09–1.16%, indicating that the UP-M-rtPCR method had good reproducibility. Second, the relative LOD was tested in 5–0.005% (*w*/*w*) binary meat mixtures for each target species. Significant fluorescence for cat, dog, fox, or mink DNA was clearly displayed even in 0.05% of the target species in a mixed background (Figure 5A). The R^2^ of each standard curve was greater than 0.99, indicating a good linear relationship (Figure 5B). Moreover, the measured Cq values in the three replicates were very similar (SD was 0.03–0.25, CV was 0.09–0.74%) (Appendix A), proving that UP-M-rtPCR had excellent performance in terms of stability and reproducibility. Generally, 1% of unspecified meat in a sample is considered the threshold for intentional adulteration or gross negligence [43]. A lower proportion of adulterants had almost no additional economic benefits for illegal meat producers. In this study, as little as 0.05 ng of DNA per reaction or 0.05% of the meat from the target species, about 19 copies for carnivorous animals, could be detected, which fully meets the European Network of GMO Laboratories (ENGL) requirements [42] of food control agencies for the detection of cat-, dog-, fox-, and mink-derived ingredients.

### 3.5. Quantitative Analysis of Cat, Dog, Fox, and Mink Fractions in DNA Mixtures Using UP-M-rtPCR

Ingredient quantification allows for the identification of trace contamination or voluntary addition. In this study, standard curves involving the relationship between Cq values and the common logarithm of different DNA proportions were generated for the quantification of cat, dog, fox, and mink fractions (Figure 6). The R^2^ of each standard curve was greater than 0.99, and the amplification efficiencies were 94.99–102.43%, meeting the 90–110% acceptance criterion of ENGL [42]. To evaluate the accuracy and precision of UP-M-rtPCR, three DNA mixtures were prepared by mixing different proportions of cat, dog, fox, and mink DNA with cattle and sheep DNA for quantitative analysis. The quantitative results for the three DNA mixtures are summarised in Table 2. The relative deviation (R.D.) ranged from 0.48% to 7.04%, while the relative standard deviation (R.S.D.) ranged from 1.42% to 13.78%. All R.D. and R.S.D. values were well within the acceptance criterion of ≤25% specified by ENGL [42], indicating that UP-M-rtPCR had good accuracy and precision for the quantitative detection of DNA proportions from cat, dog, fox, and mink in DNA mixtures. In general, owing to differences in tissue composition, meat texture, DNA degradation, and genome size of different species, quantifying the species-derived ingredients based on meat weight is only useful if the origin, composition, and treatment of the sample are already known, which is not normally the case in regulatory control [30,44]. Therefore, quantitative verification of the UP-M-rtPCR method was based on a DNA mixture with a known proportion rather than the meat mixture based on the weight ratio.

### 3.6. Application of UP-M-rtPCR in Detecting Artificial Meat Samples

To test the applicability and accuracy of this method, six artificial meat samples with the meat of multiple target species were analysed using UP-M-rtPCR. As shown in Figure 7, all target species’ components incorporated in each artificial meat sample were detected simultaneously in one reaction. Even for an artificial meat sample containing only 1% of the four target species (sample 5), four independent fluorescence amplification curves were observed simultaneously, indicating that our method behaved well in a complex meat matrix. Moreover, the cattle- and sheep-derived ingredients in each artificial meat sample were also detected using previous methods [45,46]. All the test results for the artificial meat samples are presented in Table 3. The test results of the six artificial meat samples were consistent with the actual ingredients, suggesting 100% accuracy in the detection of cat, dog, fox, and mink fractions in artificial meat samples.

The use of universal primers can reduce the competition between primers and templates and substantially improve amplification sensitivity [27,47]. Compared to conventional multiplex PCR [11,16], our UP-M-rtPCR method based on universal primers can simplify the reaction system, improve detection efficiency, and ensure identical AE for different targets, which is suitable for routine detection of meat adulteration. Furthermore, the UP-M-rtPCR method showed good performance in relation to sensitivity (0.05 ng or 0.05% *w*/*w*), even higher than that of a TaqMan multiplex real-time PCR assay targeting mitochondrial DNA (0.5% *w*/*w*) [48]. The R.D. (0.48–7.04%) and R.S.D. (1.42–13.78%) in the analysis of the three DNA mixtures indicate that our method can accurately quantify the proportion of target species DNA in unknown samples. Moreover, our method can also be used in combination with other species-specific primers and probes and shows excellent application in the detection of multi-species mixed meat products (Table 3 and Figure 7). In brief, our study developed a powerful approach for the specific identification and quantitation of cat, dog, fox, and mink DNA in one reaction, which showed excellent accuracy, sensitivity, and adaptability for the analysis of complex meat products. This is expected to provide an efficient solution for the supervision of meat products in markets. The UP-M-rtPCR method also provides inspiration for the simultaneous detection of other meat species and their products, which is believed to have wide application prospects in the future. Considering that food processing technology can affect the performance of the method, other types of samples need to be further tested by researchers.

## 4. Conclusions

Some new types of meat adulteration have been emerging in recent years; in particular, the addition of multi-species ingredients in food has greatly increased the difficulty of detection. In this study, a set of nuclear DNA sequences, universal primers, and species-specific probes were identified and evaluated for specific detection of cat, dog, fox, and mink content. Then, a simple, accurate, and efficient UP-M-rtPCR approach was developed for the identification and quantitation of cat, dog, fox, and mink fractions in meat, with a 0.05 ng absolute LOD and 0.05% relative LOD. This method simplifies the PCR system and improves the efficiency of quantitative analysis of multiple target species, showing good accuracy (R.D. from 0.48% to 7.04%), precision (R.S.D. from 1.42% to 13.78%), and adaptability. In conclusion, the UP-M-rtPCR method is a promising system for application in the field of fraud control and provides technical support for regulating the meat market and protecting public health.

## Figures and Tables

**Figure 1 foods-12-00594-f001:**
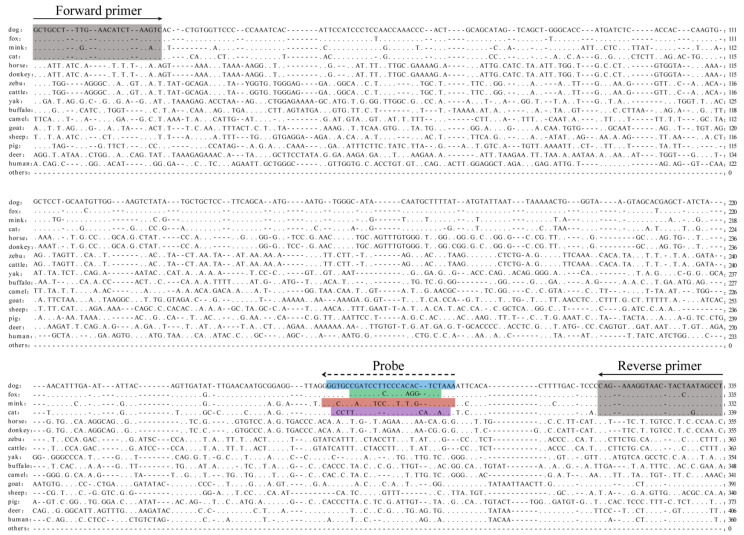
ClustalW multiple sequence alignment of the nuclear DNA target sequences against 44 target and non-target species. The gray shaded areas are the primer-binding regions. The colored shaded areas are the probe-binding regions of cats, dogs, foxes and minks.

**Figure 2 foods-12-00594-f002:**
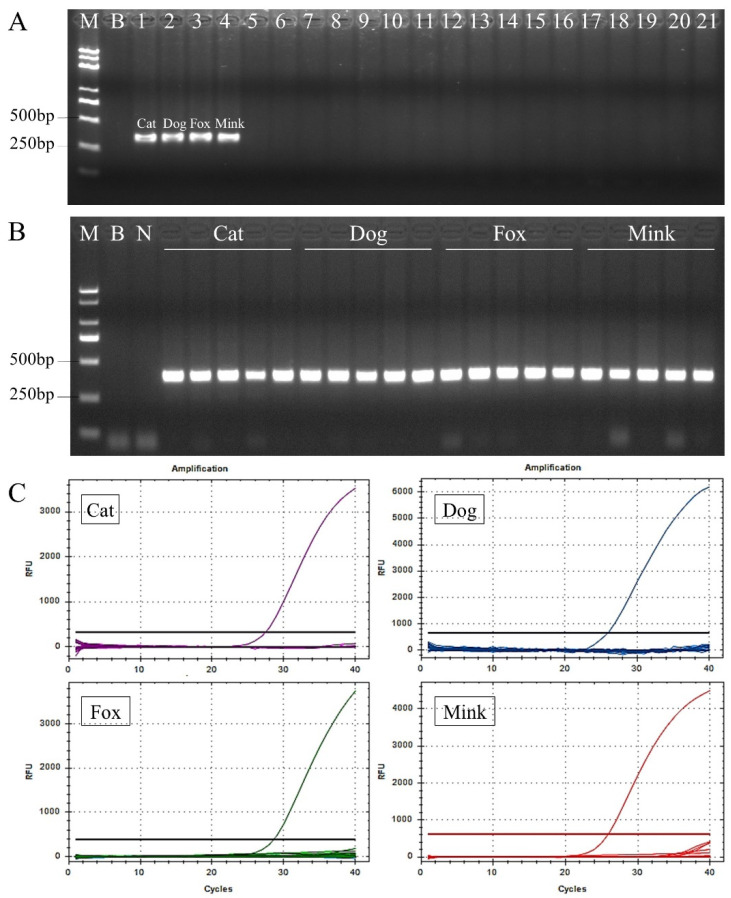
Evaluation of specificity and conservation of the nuclear DNA target sequences, primers, and probes for cat, dog, fox, and mink DNA. (**A**) Evaluation of specificity of the nuclear DNA target sequences and primers using conventional PCR. DNA templates of the conventional PCR are Lane B, blank (water); Lane N, negative control; Lane 1, cat; Lane 2, dog; Lane 3, fox; Lane 4, mink; Lane 5, domestic horse; Lane 6, donkey; Lane 7, yak; Lane 8, cattle; Lane 9, buffalo; Lane 10, sheep; Lane 11, goat; Lane 12, pig; Lane 13, deer; Lane 14, rabbit; Lane 15, mouse; Lane 16, rat; Lane 17, hamster; Lane 18, guinea pig; Lane 19, chicken; Lane 20, duck; Lane 21, goose. (**B**) Evaluation of conservation of the nuclear DNA target sequences and primers in five individual samples from each species. (**C**) Evaluation of specificity of each probe for cats, dogs, foxes, and minks using TaqMan real-time PCR.

**Figure 3 foods-12-00594-f003:**
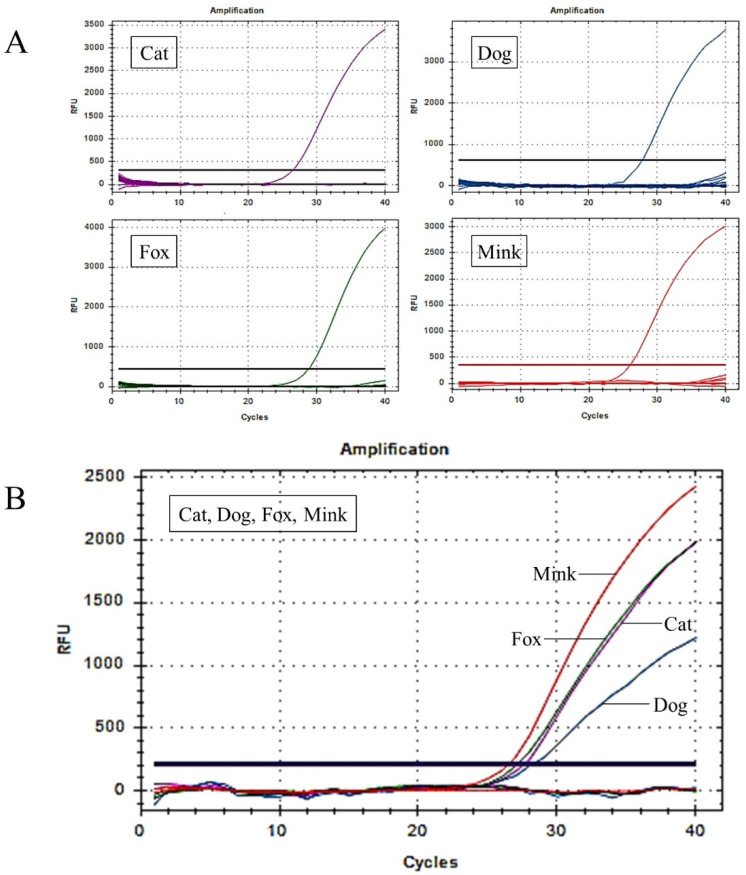
Establishment and optimisation of the UP-M-rtPCR system. (**A**) Specificity analysis of the UP-M-rtPCR system in 21 animal species. (**B**) Simultaneous detection of cat, dog, fox, and mink DNA in a single reaction using UP-M-rtPCR.

**Figure 4 foods-12-00594-f004:**
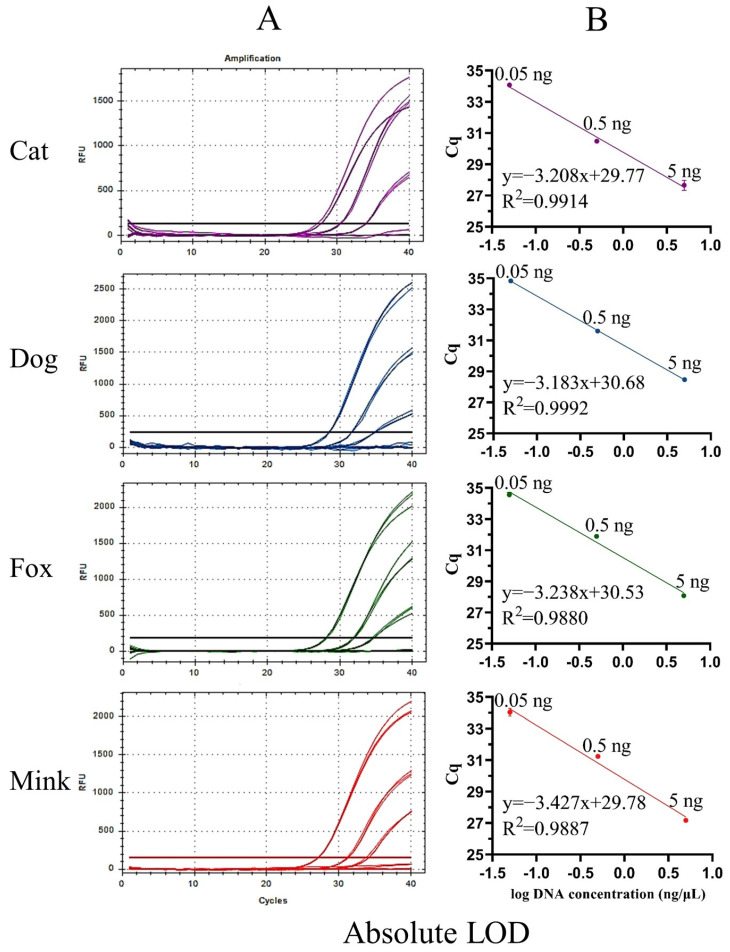
Amplification curves (**A**) and standard curves (**B**) of UP-M-rtPCR using serially diluted (10-fold) DNA (5–0.005 ng).

**Figure 5 foods-12-00594-f005:**
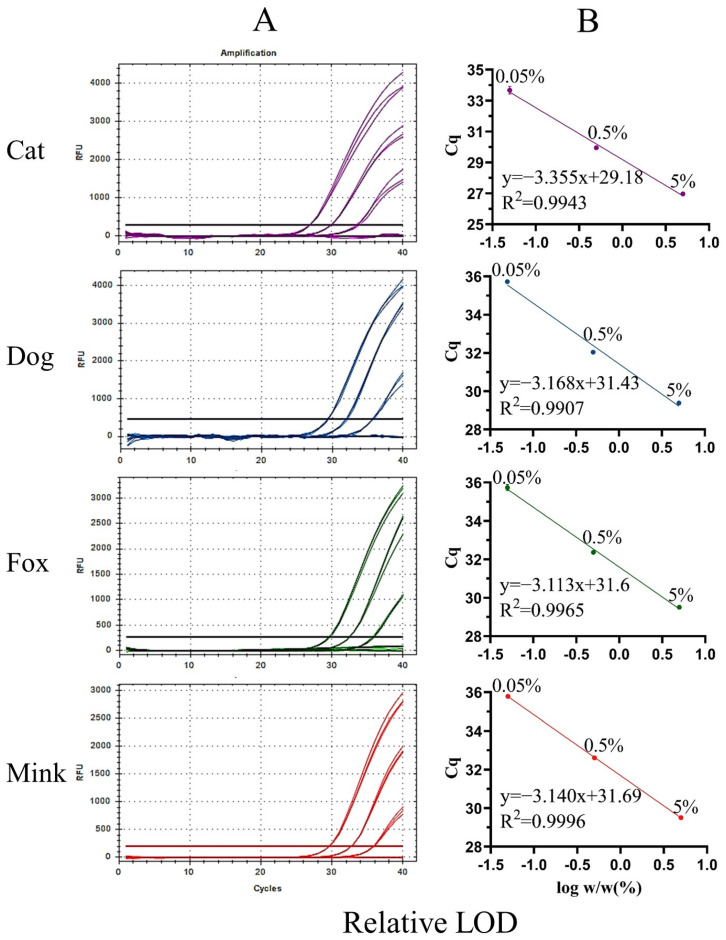
Amplification curves (**A**) and standard curves (**B**) of UP-M-rtPCR for the detection of binary meat mixtures with different proportions of cat, dog, fox, or mink meat (5%, 0.5%, 0.05%, and 0.005%).

**Figure 6 foods-12-00594-f006:**
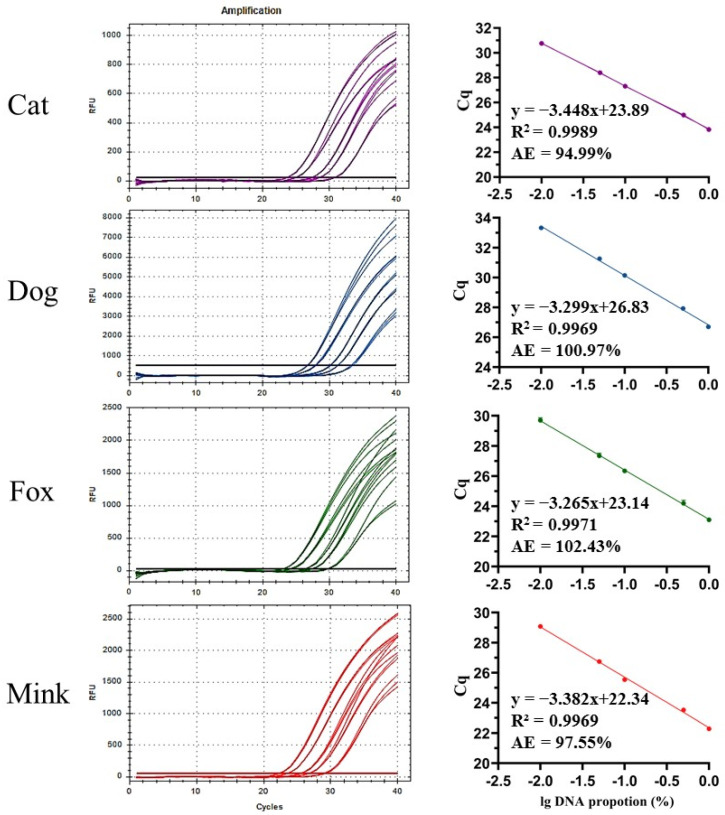
Standard curves of UP-M-rtPCR for quantitative analysis of varying DNA proportions of cat, dog, fox, or mink (1%, 5%, 10%, 50%, and 100%).

**Figure 7 foods-12-00594-f007:**
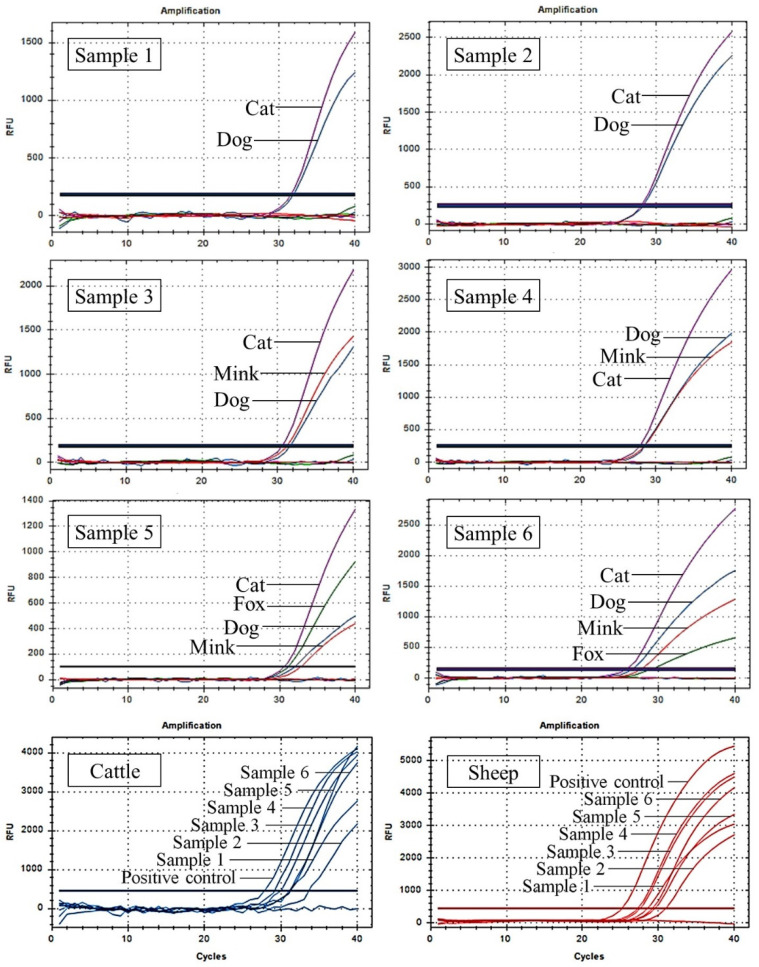
Detection of cat-, dog-, fox-, mink-, cattle-, and sheep-derived ingredients in six artificial meat samples using UP-M-rtPCR.

**Table 1 foods-12-00594-t001:** Specifications of primers and probes.

Species	Designation	Oligo Sequences (5′ → 3′)	Size (bp)
	Forward primer	GCTGCCTTK^a^GAACATCTAAR^a^TC	—
	Reverse primer	AGGCTAY^a^TR^a^GTAGTTACCTTTCTG	
Cat	Cat-P	Cy5-TTTGATGGTGTGGGGGATCAAGGC-BHQ2	339
Dog	Dog-P	FAM-TTTAGAGTGTGGGAAGGATCGGCACC-TAMRA	335
Fox	Fox-P	HEX-AGACCTTGGGGAGGATCG-MGB	335
Mink	Mink-P	Cy5.5-TTTAGACTATAGGGGAGATTGGCGCCC-BHQ3	332

^a^ Degenerate bases: K = G/T, R = A/G, Y = C/T.

**Table 2 foods-12-00594-t002:** UP-M-rtPCR quantification of cat, dog, fox, and mink content in DNA mixtures.

DNA Mixtures	Species	Actual Proportion (%)	Detected Proportion (%)	Accuracy (R.D.)%	Precision (R.S.D.)%
Ⅰ	Cat	20%	20.87%	4.36%	5.90%
	Dog	20%	18.75%	6.26%	5.58%
	Fox	20%	20.94%	4.68%	11.54%
	Mink	20%	20.37%	1.83%	7.48%
	Cattle	20%	-	-	-
Ⅱ	Cat	10%	9.66%	3.44%	4.21%
	Dog	10%	9.34%	6.58%	2.13%
	Fox	10%	10.15%	1.45%	13.83%
	Mink	10%	9.91%	0.93%	4.44%
	Sheep	60%	-	-	-
Ⅲ	Cat	5%	4.65%	7.04%	7.41%
	Dog	15%	15.59%	3.91%	2.14%
	Fox	25%	24.79%	0.85%	6.91%
	Mink	25%	24.88%	0.48%	1.42%
	Cattle	15%	-	-	-
	Sheep	15%	-	-	-

**Table 3 foods-12-00594-t003:** Analysis of six artificial meat samples.

Artificial Meat Samples.	Actual Ingredient and Proportion	Detected Species						PCR Consistency
		Cat	Dog	Fox	Mink	Cattle	Sheep	
Sample 1	cat (1%), dog (1%), cattle (49%), sheep (49%)	√	√	-	-	√	√	√
Sample 2	cat (25%), dog (25%), cattle (25%), sheep (25%)	√	√	-	-	√	√	√
Sample 3	cat (1%), dog (1%), mink (1%), cattle (48.5%), sheep (48.5%)	√	√	-	√	√	√	√
Sample 4	cat (20%), dog (20%), mink (20%), cattle (20%), sheep (20%)	√	√	-	√	√	√	√
Sample 5	cat (1%), dog (1%), fox (1%), mink (1%), cattle (48%), sheep (48%)	√	√	√	√	√	√	√
Sample 6	cat (16.67%), dog (16.67%), fox (16.67%), mink (16.67%), cattle (16.67%), sheep (16.67%)	√	√	√	√	√	√	√

## Data Availability

Data are contained within the article or Appendix A.

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
