# Peer review of "A Novel Universal Primer Multiplex Real-Time PCR (UP-M-rtPCR) Approach for Specific Identification and Quantitation of Cat, Dog, Fox, and Mink Fractions Using Nuclear DNA Sequences"

_foods, 2023, doi:10.3390/foods12030594_

Round 1

Reviewer 1 Report

The manuscript presented by Wang et al. Shows the development of a detection methodology to identify and quantify the adulteration of meat with cat, dog, fox and mink meat. The design of the study was well conducted, and correct evaluation was performed. Overall high scientific quality was demonstrated in this study and the a advise  with minor revision.

A few comments a raised below to clarify some statements and improve the quality of the manuscript

Authors need to double check all manuscript as is observed differences in the size of letters for instance in line 22-23, 123,225-268

Line 59: change “junior research” to non-experienced researcher or something similar

Line 105: Need to detail the protocol for the DNA extraction. It’s an important step that can influence the detection method

Line 299-305. In this statement you should compare results with current regulation in place and cite these regulations

In figure 6. You should indicate what AE stand and refer this abbreviation in the text

Line 324 you should also state what it’s the ENGL before mentioning it

Table 3. you should resize the size of the columns to do not have words cut

In the last part of you results you should also mention that other type of samples needs to be tested, as more processed food can affect the performance of the method.

Reviewer 2 Report

Dear authors,

The comments are attached.

regards
